# Structure of the turnover-ready state of an ancestral respiratory complex I

Bozhidar S. Ivanov [1], Hannah R. Bridges [1,2], Owen D. Jarman[1,3] & Judy Hirst [1] ✉

Respiratory complex I is pivotal for cellular energy conversion, harnessing energy from NADH:ubiquinone oxidoreduction to drive protons across energy-transducing membranes for ATP synthesis. Despite detailed structural information on complex I, its mechanism of catalysis remains elusive due to lack of accompanying functional data for comprehensive structure-function analyses. Here, we present the 2.3-Å resolution structure of complex I from the α-proteobacterium *Paracoccus denitrificans*, a close relative of the mitochondrial progenitor, in phospholipid-bilayer nanodiscs. Three eukaryotic-type supernumerary subunits (NDUFS4, NDUFS6 and NDUFA12) plus a novel L-isoaspartyl-O-methyltransferase are bound to the core complex. Importantly, the enzyme is in a single, homogeneous resting state that matches the closed, turnover-ready (active) state of mammalian complex I. Our structure reveals the elements that stabilise the closed state and completes *P. denitrificans* complex I as a unified platform for combining structure, function and genetics in mechanistic studies.

Respiratory complex I (NADH:ubiquinone oxidoreductase, CI) is an essential metabolic enzyme in mammalian mitochondria and central to oxidative phosphorylation in many aerobic organisms[1,2]. Complex I transfers two electrons from NADH to ubiquinone, feeding electrons to the electron transport chain for $O_2$ reduction, and captures the free energy released to transfer four protons[3] across an energy-transducing membrane, generating the proton motive force that powers ATP synthesis. In mammalian cells it is essential for keeping the $NAD^+$ pool oxidised for metabolism and a key contributor to reactive oxygen species production under challenging conditions[4]. Thus, complex I dysfunctions both initiate and escalate pathological mechanisms in a wide range of mitochondrial, metabolic and neuromuscular disorders[5,6].

It is widely accepted that mitochondria originated in an endosymbiotic event in which an α-proteobacterium was engulfed by an archaeon[7,8] then became reliant on its host for substrates, metabolites, and imported proteins, retaining only a small genome and reciprocating by performing $O_2$ reduction to drive ATP production. The α-

proteobacterium *Paracoccus denitrificans* is a free-living descendent of the mitochondrial progenitor that has retained many features in common with modern mitochondria[9,10]. It is also a powerful model system for studies of mammalian complex I, overcoming experimental hurdles that restrict studies of the mammalian form itself[11,12]. All species of complex I contain 14 conserved core subunits (or their equivalent) organised into two seven-subunit domains: a hydrophilic domain that catalyses the redox reaction, and a membrane domain for proton pumping. Mammalian complex I has evolved additional complexity in the form of 31 supernumerary subunits, required for assembly, stability and regulation[13,14], and its seven membrane-bound core subunits, being encoded in the mitochondrial genome, are refractory to mutagenesis. In contrast, bacterial model systems contain minimal, simpler complexes I in which all the subunits can be readily mutated[11,15,16]. The core subunits of human complex I share higher sequence similarity to complex I from *P. denitrificans* (*Pd*-CI) than to other bacterial models such as *Escherichia coli* or *Thermus thermophilus*[17]. Furthermore, like the mammalian enzyme, but unlike *E.*

[1]The Medical Research Council Mitochondrial Biology Unit, University of Cambridge, Keith Peters Building, Cambridge Biomedical Campus, Cambridge, UK. [2]Present address: Structura Biotechnology Inc., Toronto, Canada. [3]Present address: Department of Biochemistry and Synthetic Metabolism, Max Planck Institute for Terrestrial Microbiology, Marburg, Germany. ✉e-mail: jh480@cam.ac.uk

*coli* or *T. thermophilus*, *Pd*-CI uses ubiquinone-10 as its sole electron acceptor[9,10], has been demonstrated to exhibit a $4H^+$/NADH proton-pumping stoichiometry[3], catalyses reverse electron transfer[11,18], resides in supercomplex assemblies[19], and contains a minimal complement of mammalian-like supernumerary subunits (NDUFS4, NDUFS6 and NDUFA12)[17].

Despite numerous high-resolution electron cryomicroscopy (cryo-EM) structures of CI from various species, there is no consensus on the coupling mechanism that captures the free energy released from the redox reaction to drive proton pumping, or on the proton pumping mechanisms or transfer pathways. To advance knowledge of these mechanisms it is necessary to combine structural investigations with biophysical, biochemical and genetic strategies. Alongside biochemical studies on the purified enzyme, the ability of *P. denitrificans* to form tightly coupled sub-bacterial particles (SBPs) enables powerful functional investigations of *Pd*-CI to be combined with point mutations throughout the enzyme, including in the proton-pumping subunits encoded by the mitochondrial genome in eukaryotes. Here, we develop a robust approach for single-particle cryo-EM on *Pd*-CI in phospholipid nanodiscs and report the structure in its turnover-ready resting state at 2.3-Å resolution. We thereby add the final element to our toolkit of structure, biophysics, biochemistry and genetics in a single system to realise the power of *Pd*-CI for investigations of complex I catalysis.

## Results

### Cryo-EM analysis of *P. denitrificans* complex I in DDM (*Pd*-CI-DDM)

Highly catalytically active *Pd*-CI was purified using dodecylmaltoside (DDM)[12], frozen directly onto PEGylated gold grids and imaged with an FEI Titan Krios microscope and K2 detector (Supplementary Table 1). Although classes of particles lacking clear density for the distal section of the membrane domain were discarded, the final 4.1-Å resolution reconstruction of 51,308 particles still exhibited poor density in this region (Supplementary Fig. 1a). Subsequently, 3D variability analysis (3DVA) uncovered flexibility in the intact membrane domain and allowed separation into two discrete classes (Supplementary Figs. 1a and 2a, b). From class 1 (19,736 particles) to class 2 (31,572 particles) the distal subdomain, comprising subunits ND4 (Nqo13, see Supplementary Table 2 for subunit nomenclature) and ND5 (Nqo12), moves downwards and sideways such that the class 2 membrane domain appears straighter (from the top) and flatter (from the side) (Supplementary Fig. 1b). The resulting cleft between the distal and proximal subdomains resembles that observed previously in the (likely inactive) slack state of bovine complex I[20]. Since the distal subdomain does not remain firmly attached to the complex, we chose to next optimise the stability of *Pd*-CI for improved cryo-EM grid preparation.

### Reconstitution of *P. denitrificans* complex I into nanodiscs (*Pd*-CI-ND)

To stabilise the distal subdomain and minimise detergent artefacts, we investigated phospholipid nanodiscs as a membrane mimetic system. *Pd*-CI-DDM was reconstituted into nanodiscs containing phosphatidylcholine, phosphatidylethanolamine, cardiolipin and ubiquinone-10 using membrane scaffold protein (MSP) MSP2N2[20,21], and the stability and activity of *Pd*-CI-DDM and *Pd*-CI-ND compared. First, in nano-differential scanning fluorimetry (nano-DSF) experiments (Supplementary Fig. 3) the fluorescence is recorded as the temperature is increased, and the two peaks observed in the first derivative were previously ascribed to unfolding of first the membrane domain and then the hydrophilic domain[11,22]. Whereas glycerol, NaCl and $CaCl_2$ were found to modulate the stability of *Pd*-CI-DDM (Supplementary Fig. 3a–c), significantly higher stability was observed for *Pd*-CI-ND (Supplementary Fig. 3d–f) and highlighted by a direct comparison of matching solution conditions (Supplementary Fig. 3c, e). Second, the

ability of *Pd*-CI-ND to catalyse NADH:ubiquinone oxidoreduction was tested using either the ubiquinone-10 ($Q_{10}$) present in the nanodiscs, or decylubiquinone (dQ), a short-chain ubiquinone that can be added to assays in much higher amounts (Supplementary Fig. 4). NADH:$Q_{10}$ rates of ~8 µmol NADH $min^{-1}$ $mg^{-1}$ (76 $s^{-1}$) were observed, dependent on reoxidation of any ubiquinol formed by an alternative ubiquinol oxidase (AOX)[23] (Supplementary Fig. 4a, c). Similar rates observed with dQ (~9 µmol $min^{-1}$ $mg^{-1}$) (85 $s^{-1}$) were approximately doubled (recovering the rate of *Pd*-CI-DDM[12]) by addition of either AOX, or a mixture of asolectin and CHAPS to dissociate the MSPs and solubilise the nanodiscs (Supplementary Fig. 4b, d). These high, inhibitor-sensitive rates demonstrate the catalytic ability of *Pd*-CI-ND, but also suggest the MSPs hinder ubiquinone exchange. Ubiquinone/ubiquinol exchange by mammalian complex I in MSP2N2 nanodiscs was previously found to be blocked by the MSPs[20].

### Cryo-EM analyses of *Pd*-CI-ND

Two matching preparations of *Pd*-CI-ND were frozen onto PEGylated gold grids and imaged using an FEI Titan Krios microscope and K3 detector (Table 1 and Supplementary Figs. 5 and 6). The second preparation was augmented by 0.01% fluorinated octyl maltoside (FOM, confirmed not to affect *Pd*-CI-ND stability in Supplementary Fig. 3f), aiming to boost the resolution via improved particle distribution and reduced ice thickness[24]. During processing, relatively poor map quality was observed for the distal membrane subdomain, and classes that lacked density for it were again identified suggesting the subdomain was lost prior to reconstitution. Both datasets yielded maps for intact *Pd*-CI-ND with global resolutions of 2.4–2.5 Å (Supplementary Figs. 5 and 6), and as no substantial differences could be identified between them, they were combined (Supplementary Fig. 7) into a final 2.3-Å resolution reconstruction (Supplementary Fig. 2c and Table 1). Different membrane domain conformations could not be separated, but the final map still exhibited lower resolution for the distal subdomain (Supplementary Fig. 2c). Focussed refinement on this region provided a 2.5-Å resolution map (Supplementary Figs. 2d and 7 and Table 1) to assist model building.

Inspection of the maps revealed fragmented densities on top of the distal membrane subdomain and on one side of the nanodisc. The particles were then separated into two classes on this basis, one with the nanodisc tight around the membrane domain and the other with an extended disc density (Supplementary Fig. 8). The latter may represent a phospholipid/detergent micelle that has partially displaced the MSPs, relaxing their blockade of ubiquinone exchange and explaining the ability of our preparation to catalyse ubiquinone reduction (Supplementary Fig. 4). Further classification showed the density above the distal subdomain was present in only a portion of the tight-nanodisc class, and local refinement allowed it to be identified as *P. denitrificans* cytochrome $bc_1$ complex (complex III, PDB: 2YIU)[25] (Supplementary Fig. 8). However, the interaction is unambiguously artefactual, as the periplasmic *Pd*-CIII Rieske protein/cytochrome $c_1$ is bound to the cytoplasmic side of *Pd*-CI subunit ND5.

A de novo model of the 2.3 Å-resolution map of *Pd*-CI-ND (Supplementary Fig. 7) was created using ModelAngelo[26] and the sequences of 17 known *Pd*-CI subunits[12,17] including a further, newly-identified subunit, a protein L-isoaspartyl-O-methyltransferase (PIMT, referred to as CI-PIMT). An unassigned density on the hydrophilic domain was readily identified as CI-PIMT by assessing candidate proteins from mass spectrometry analyses of *Pd*-CI-DDM[12] and their homologous structures in the Protein Data Bank. Fig. 1 shows an overview of the final refined structure containing 97% of the residues (Supplementary Table 2) and 1080 waters (Table 1).

### Structural similarity of *Pd*-CI and mammalian complex I

The seven membrane-bound core subunits of *Pd*-CI, encoded in the mitochondrial genome in mammals, have 23-39% sequence identity

**Table 1 | Cryo-EM data collection, refinement, and validation statistics for *Pd*-CI-ND**

| Data collection and processing | *Pd*-CI-NDs | |
|---|---|---|
| Voltage (kV) | 300 | |
| Nominal magnification | 81,000× | |
| Electron exposure (e⁻ Å⁻²) | 46 [grids 1+2], 45 (grid 3) | |
| Defocus range (μm) | −1.0 to −2.4 [grids 1+2], −0.9 to −2.5 (grid 3) | |
| Calibrated pixel size (Å) | 1.066 [grids 1+2], 1.070 (grid 3) | |
| Number of frames | 40 | |
| Symmetry imposed | C1 | |
| Number of micrographs | 11,994 (grid 1), 4820 (grid 2), 4561 (grid 3) | |
| Initial particle images (no.) | 1,034,504 [grids 1 + 2], 320,429 (grid 3) | |
| Final particle number (no.) | 103,186 [grids 1 + 2], 43,417 (grid 3) | |
| | Consensus *Pd*-CI-ND map (EMDB: 18324) | ND4-5 focus refined map (EMDB: 18325) |
| Map sharpening *B* factor (Å²) | −39.40 | −48.88 |
| Map angular accuracy (°) | 0.234 | 0.468 |
| Map translational accuracy (Å) | 0.215 | 0.540 |
| Final map sampling | 0.745 | 0.745 |
| Map resolution (Å) (FSC = 0.143) | 2.3 (0.143) | 2.5 (0.143) |
| Map resolution range (Å) | 2.2–4.8 | 2.4–4.0 |
| Refinement | | |
| Initial model used | de novo (ModelAngelo) | *Pd*-CI-ND |
| Model resolution (Å) (FSC = 0.5) | 2.4 | 2.6 |
| Model composition | | |
| Nonhydrogen atoms | 84,464 | 18,950 |
| Protein residues | 5135 | 1118 |
| Ligands | 49 | 15 |
| Waters | 1080 | 115 |
| *B* factors mean (Å²) | | |
| Protein | 39.58 | 46.61 |
| Ligand | 56.03 | 62.48 |
| Water | 38.92 | 48.17 |
| RMS deviations | | |
| Bond lengths (Å) | 0.006 | 0.004 |
| Bond angles (°) | 0.612 | 0.557 |
| Validation | | |
| MolProbity score | 1.15 | 1.10 |
| EMRinger score | 5.74 | 5.61 |
| Clashscore | 3.04 | 2.92 |
| Rotamer outliers (%) | 0.36 | 0.45 |
| Cβ outliers (%) | 0.00 | 0.00 |
| Ramachandran plot | | |
| Favoured (%) | 97.74 | 97.93 |
| Allowed (%) | 2.24 | 2.07 |
| Outliers (%) | 0.02 | 0.00 |
| Rama-Z (Ramachandran *Z*-score, RMSD) | | |
| Whole (*N* = 5091) | 0.26 (0.11) | 0.76 (0.25) |
| Helix (*N* = 2728) | 0.60 (0.10) | 0.98 (0.18) |
| Sheet (*N* = 420) | 0.31 (0.25) | 1.18 (1.19) |
| Loop (*N* = 1943) | -0.30 (0.14) | -0.74 (0.38) |

with the subunits from bovine complex I, our mammalian reference enzyme (Supplementary Table 3). Comparison of individual subunit structures revealed mostly matching positions for the transmembrane helices (TMHs) that dominate the domain (Fig. 2), although *Pd*-ND2 (Nqo14) contains the three N-terminal TMHs lost from the mammalian complex during evolution[27] (Fig. 3a) and TMH4 of ND6 (Nqo10) is shifted (Fig. 3b). In mammalian complex I ND6-TMH4 is next to ND6-TMH1[13,20,28], but in *Pd*-CI, as in CI from other bacteria[15,29–31] and single cell eukaryotes[32,33], it is located towards ND5-TMH16. The *Pd*-ND6 TMH3-4 and TMH4-5 loops vary to accommodate the shift, and the TMH3-4 loop is not fully resolved. Finally, a substantial 33-residue C-terminal extension of *Pd*-ND6 reaches up onto the hydrophilic domain, replacing the N-terminus of mammalian NDUFS8 (Nqo9) in complementing a β-strand in NDUFS2 (Nqo4, residues 265-8) and likely helping to stabilise the domain interface (Fig. 3b).

On the periplasmic side of *Pd*-CI, as in other bacteria[15,29,31] and yeast[32], the ND4 TMH2-3 β-hairpin loop (β1-2) is extended relative to in mammalian species[13,20,28] and a corresponding β-hairpin is found in the ND2 TMH2-3 loop, stabilising the domain in the absence of mammalian subunits NDUFA8 and NDUFS5 (Fig. 3a). Two hitherto-unobserved periplasmic β-hairpins (β3-4) are present in the *Pd*-ND4 and ND5 TMH6-7 loops, and the ND5-TMH14-15 loop contains a substantial flexible periplasmic insertion (Fig. 3a). The insertion includes 44 disordered residues that are predicted with 'very low' confidence by AlphaFold[34]. Additionally, the intrahelical loop in *Pd*-ND5 TMH12, which protrudes into the membrane and dips towards the periplasmic side, is unusually large (11 *vs.* 5 residues in other structures) (Fig. 3a). Finally, two Ca²⁺ are observed on the periplasmic ND2 and ND4 surfaces (Fig. 3a) coordinated by the ND2-Asp260 carboxylate and L477 carbonyl, and the ND4-L197 and P275 carbonyls, but with unresolved densities for some of the coordinating waters. As CaCl₂ was present in the buffer and the Ca²⁺ do not occupy known functional regions they may be bound adventitiously.

Core subunits in the *Pd*-CI hydrophilic domain show higher sequence identities with the mammalian subunits at 40-73% (Supplementary Table 3), perhaps reflecting faster divergent evolution of the mammalian mitochondrial genome[35]. NDUFV1 (Nqo1) in the NADH-dehydrogenase module, containing the NADH-binding active site and one [4Fe-4S] cluster, is very highly conserved structurally (Supplementary Table 3). The FeS-domain of NDUFS1 (Nqo3), containing two [4Fe-4S] and one [2Fe-2S] clusters, also matches well, while subunit NDUFV2 (Nqo2) and the C-terminal domain of NDUFS1, which are peripheral to electron transfer, match less closely (Fig. 2). Notably, the *Pd*-NDUFS1 C-terminal domain does not contain the extra [4Fe-4S] cluster found in *E. coli* and other bacteria[15,29–31] and its structure matches the mammalian (not the *E. coli*) form[13,20,28] (Supplementary Table 3). Intriguingly, the *Pd*-NDUFV2 C-terminus is 29 residues longer than in mammalian NDUFV2 and partially overlays a small animal-specific supernumerary subunit, NDUFV3 (Fig. 3c). Lower down the hydrophilic domain, NDUFS2, NDUFS3 (Nqo5), NDUFS7 (Nqo6), and NDUFS8 are highly structurally conserved with mammalian species, except for the NDUFS2 N-terminus (Fig. 2 and Supplementary Table 3). These subunits contain three [4Fe-4S] clusters and, together with ND1 (Nqo8), form the ubiquinone-binding site. Density for the dimethylarginine in NDUFS2[36,37], conserved in eukaryotic complex but not reported for any other bacteria, is observed at position 65. The N-terminus of NDUFS2 (Fig. 3d) is structured very differently: in mammalian complex I it extends along the matrix surface of the membrane domain[13,20,28], whereas in *Pd*-CI it turns back toward the hydrophilic domain (and in complex I from *E. coli* CI is absent entirely[15,30]).

**Mammalian-homologous supernumerary subunits of *Pd*-CI**

*P. denitrificans* supernumerary subunits NDUFS4, NDUFS6 and NDUFA12 share ~30% sequence identity with their mammalian

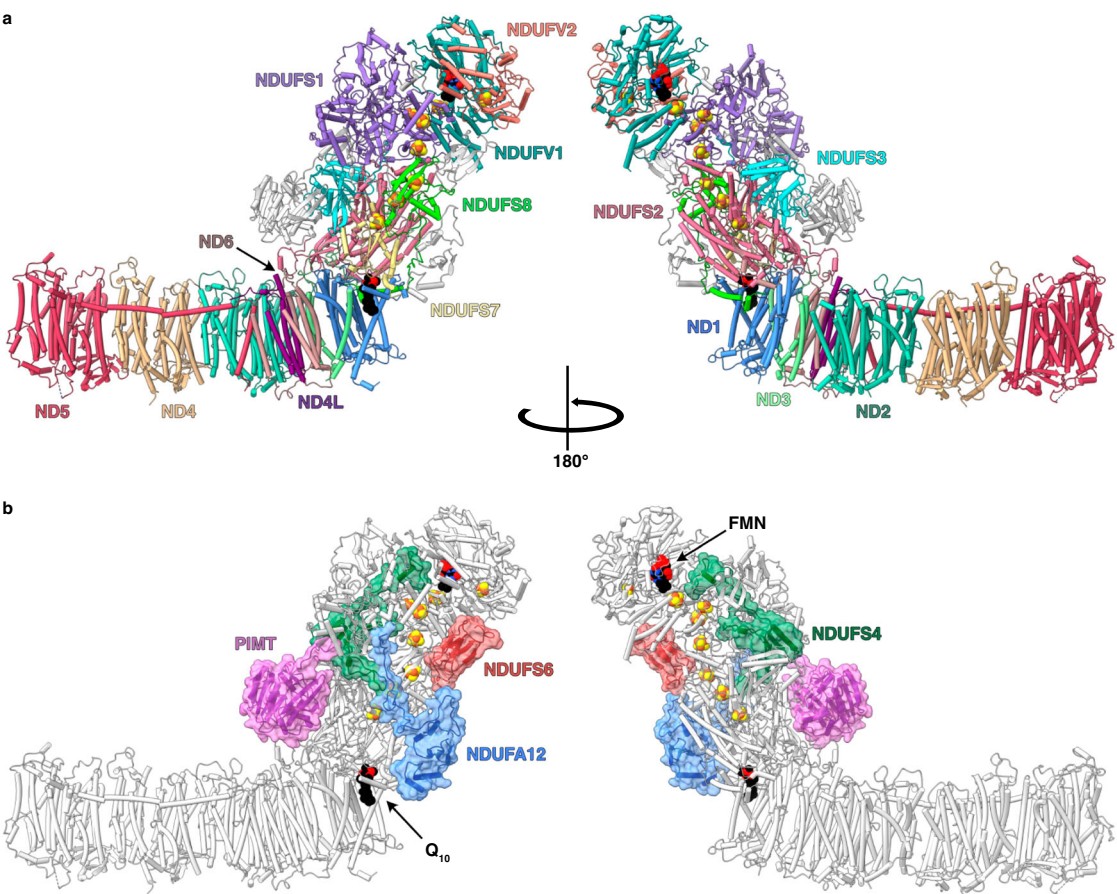

**Fig. 1 | The structure of complex I from *P. denitrificans*. a** The 14 core subunits are shown and labelled in colour with the supernumerary subunits in grey. **b** The supernumerary subunits are shown and labelled in colour with transparent surfaces, with the core subunits in grey. Subunits with homologues in human CI are named using the human nomenclature; the PIMT subunit is specific to *P. denitrificans*. FeS clusters are shown in orange and yellow.

counterparts and are well conserved structurally (Fig. 2 and Supplementary Table 3). *Pd*-NDUFS4 lacks the ~30 residue N-terminal coil of the mammalian subunit. However, the 4-strand β-sheet and α-helix, which anchor the NADH-dehydrogenase module (N-module) to NDUFS3, and the C-terminal coil that binds to the interface between NDUFS1, NDUFV1 and NDUFV2 (Fig. 4a) match closely, consistent with deletion of NDUFS4 destabilising the N-module in both mouse[38] and *Yarrowia lipolytica*[39]. *Pd*-NDUFS6 contains a conserved 4-strand β-sheet sandwich that coordinates a $Zn^{2+}$ and interacts predominantly with NDUFS1 and NDUFS8, also stabilising the N-module (Fig. 4a). An octahedrally coordinated $Ca^{2+}$ bound 11.2 Å from the $Zn^{2+}$ bridges the NDUFS6-Asp48 and NDUFS8-Asp88 carboxylates (Fig. 4b). *Pd-ndufs6* does not encode the N-terminal domain present in the mitochondrial subunit (Fig. 4b)[13,14] so, unlike in mammalian CI, *Pd*-NDUFS6 does not interact with *Pd*-NDUFS4 or *Pd*-NDUFA12 (Fig. 4b). *Pd*-NDUFA12 begins with two amphipathic α-helices, positioned differently in the mammalian complex (Fig. 4b), which are proposed to govern the membrane structure around the ubiquinone-binding site[14]. The subsequent three stranded β-sheet, α-helix and C-terminal coil mainly overlay the mammalian subunit, on the 'heel' of the complex.

## PIMT supernumerary subunit of *Pd*-CI
Protein aspartate and asparagine residues are susceptible to spontaneous degradation over time, forming species such as L-isoaspartate that may disrupt structure and function[40]. PIMT enzymes catalyse S-adenosyl-methionine (SAM) to S-adenosyl-homocysteine (SAH) dependent methylation, the first step in repair of L-isoAsp[40]. *P. denitrificans* contains two PIMT genes (A1B5L6 and A1B5M0, 29% sequence

identity) that we name CI-PIMT (Figs. 1 and 4a) and Free-PIMT. Comparing the AlphaFold Free-PIMT model with our CI-PIMT structure revealed close overlay of secondary structures (overall RMSD 3.87 Å) but opposing surface charge distributions. CI-PIMT has a large acidic patch on its outward-facing side, and a basic patch at its interface with the acidic surface of NDUFS3 that is replaced by an acidic surface on Free-PIMT (Fig. 4c). The α-proteobacterium *Rhodopseudomonas palustris* also encodes multiple PIMTs, one of which (RPA2580) is catalytically inactive[41] and exhibits similar sequence features with CI-PIMT (Supplementary Fig. 9). First, the canonical Thr60 and Ser62 residues (CI-PIMT numbering) form hydrogen bonds to the L-iso-Asp, and the S62A mutation in *Drosophila melanogaster* abolished PIMT activity[42]. In *Pd*-CI-PIMT, Thr60 and Ser62 are mutated to Val and Leu, and to Phe and Leu in RPA2580. Second, Asp134 and Gly135, which form hydrogen bonds to the SAM/SAH are mutated to Ala and Leu in CI-PIMT and Glu and Ala in RPA2580. Cofactor binding is also obstructed in CI-PIMT by Arg59, consistent with the lack of cofactor density (Fig. 4d). Finally, the mode of CI-PIMT binding to complex I obstructs access of L-isoAsp-containing proteins to its active site (Fig. 4e). We conclude CI-PIMT has diverged from the PIMT family, losing its catalytic ability, and assume the canonical Free-PIMT performs the PIMT role in *P. denitrificans* cells.

## *Pd*-CI adopts a single closed/active resting state
Cryo-EM analyses of resting preparations of mammalian CI have identified distinct 'closed' and 'open' states that correspond to the biochemically defined 'active' and 'deactive' resting states[13,37,43–45]. Whereas the active resting state is a turnover-ready state, the

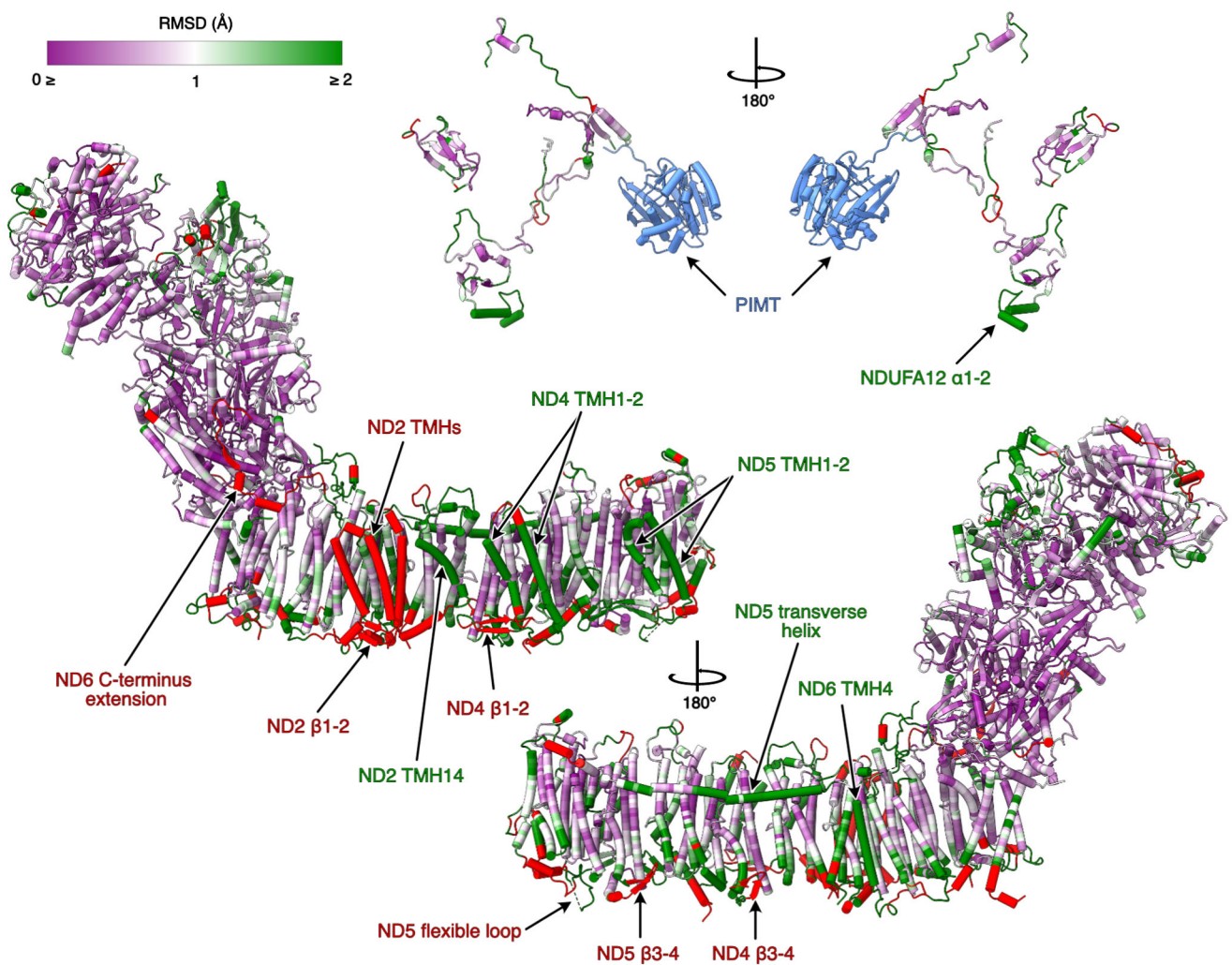

**Fig. 2 | Cα-RMSD comparison of the structures of *P. denitrificans* and *B. taurus* (PDB-7QSL[20]) complex I.** The structure of *Pd*-CI is coloured by Cα RMSD value following per-subunit alignment in UCSF ChimeraX. A purple-white-green palette from 0 to 2 Å was used to colour the structure. Unpaired regions specific to *P. denitrificans* complex I are in red and the *P. denitrificans*-specific PIMT subunit is in blue. The core and supernumerary subunits are shown separately with key elements of interest indicated. See Fig. 1 for the locations of individual subunits.

pronounced deactive resting state is dormant and requires reactivating to catalyse. In contrast, *Pd*-CI is in a single, homogenous resting state. The two mammalian states can be distinguished biochemically using N-ethyl maleimide, which reacts with ND3 (Nqo7)-Cys39 (*Pd*-Cys47) in the deactive state, where it is exposed to solvent (but not in the active state where it is buried) to prevent catalysis[46]. *Pd*-CI is insensitive to NEM[12] and displays all the structural hallmarks of the closed state[13,37,44,45] (Fig. 5a, b). Specifically, the ND3 TMH1-2, ND1 TMH5-6 and NDUFS2 β1-β2 loops at the domain interface and ubiquinone-binding channel are well ordered in their closed-state conformations (Fig. 5a), ND3-Cys47 is buried, NDUFS7-Arg77 points away from NDUFS2 adjacent to a loop element, ND1-TMH4 is bent, and ND6-TMH3 is fully α-helical (Fig. 5b). The ubiquinone-binding site is sealed from the cytosol (Fig. 5c), matching all mammalian closed/active states[13,20,28,37,44,45,47]. Therefore *Pd*-CI is in the active resting state. Finally, we note that interdomain angles cannot be used to assign states as open/closed by comparison to mammalian references because species-dependent variations dominate the more subtle opening/closing effects (Supplementary Table 4). Indeed, in the mammalian enzyme the domain motion is better characterised as a twisting or torque in which the domains rotate against each other, only giving the visual impression of an angular opening[13]. In *E. coli* complex I

it has also been noted that the different opening movement is not dominated by an angular opening[15].

We propose *Pd*-CI is stabilised in the closed state by the PIMT, which binds on top of the ND3-TMH1-2 and NDUFS2 N-terminal loops (Fig. 5d). In addition, a bound $Ca^{2+}$ (Fig. 3d), coordinated by the NDUFS2-Asn8, Asp49 and Glu54 carboxylates, Arg6 carbonyl, and one resolved water (a second is assumed), stabilises the NDUFS2 N-terminus and bridges it to the β2-β3 strands (Fig. 5d, e). Occupying the interdomain 'corner', the N-terminus also directly contacts ND1, ND6, ND3 and ND4L (Nqo11) in the membrane domain, including the ND3-TMH1-2 loop and extended ND6 N-terminus. Therefore, it appears that both the PIMT and the $Ca^{2+}$-bound NDUFS2 N-terminus stabilise the turnover-ready closed state of *Pd*-CI.

## Ubiquinone inserted partially into the binding site

The ubiquinone-binding cavities in *Pd*-CI and the active state of bovine complex I[20] are very similar, except in *Pd*-CI the E-channel branch in ND1 is narrower, and an additional lobe is present in NDUFS2 (Fig. 6a). Density in the central polar region of the channel has been assigned to a partially inserted ubiquinone-10 molecule with its headgroup within 4 Å of ND1-Arg38, Trp41, Met244, Lys295 and NDUFS7-Arg77 (Fig. 6a, b). NDUFS7-Arg77, which changes conformation upon

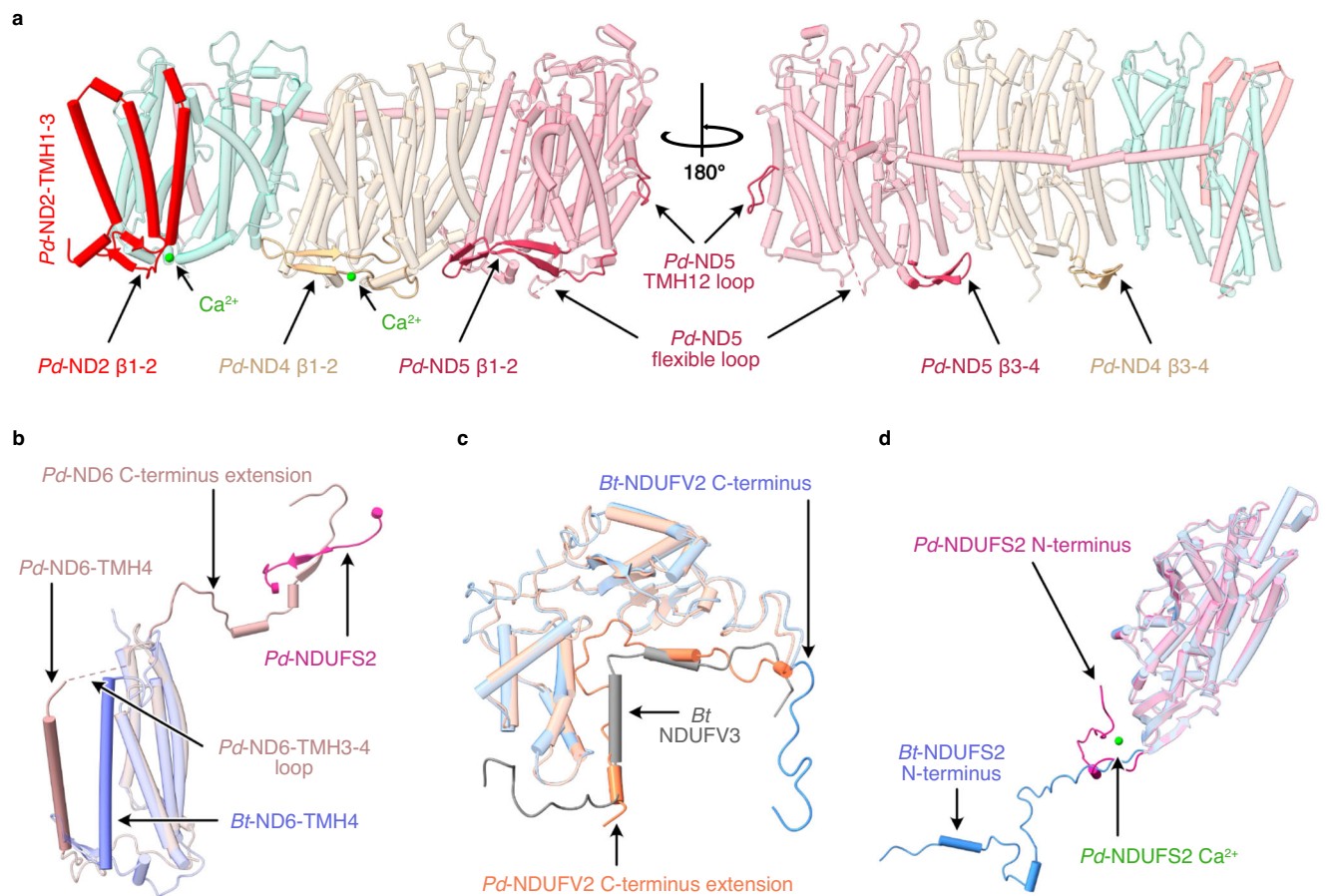

**Fig. 3 | Structural elements in the core subunits of complex I from *P. denitrificans* that differ from their mammalian homologues. a** Subunits ND2, ND4 and ND5, showing the additional N-terminal TMHs in ND2, the β1-β2 and β3-β4 loops in ND4 and ND5 and the two bound Ca²⁺ ions. **b** The altered position of ND6-TMH4 and the C-terminal extension. **c** The C-terminus of *Pd*-NDUFV2 overlapping with bovine supernumerary subunit NDUFV3. **d** The different position of *Pd*-NDUFS2 N-terminus and the Ca²⁺ bound in *Pd*-NDUFS2. The *Pd*-CI structure was compared with the bovine active-apo state (no bound ligand) PDB-7QSL[20].

complex I opening (deactivation) in mammalian species[45], has shifted to accommodate the headgroup. The headgroup location closely matches the quinones modelled previously in the closed ovine[28], *E. coli*[15] and *Mycobacterium smegmatis*[31] enzymes (Supplementary Fig. 10) and the earlier predictions from molecular simulations[48,49]. As both the enzyme and ubiquinone are oxidised, the observed binding pose is not a reactive intermediate, and the substrate is required to move further into the channel for reduction. The ubiquinone-10 has also been observed inserted ~20 Å further into the closed channel in the bovine[20] and porcine[47] enzymes, placing the headgroup adjacent to NDUFS2-His39 and Tyr88 (Supplementary Fig. 10d). His39 and Tyr88 were found here in orientations matching the bovine active-apo state[20]. The isoprenoid chain of the fully inserted molecules overlaps the bound ubiquinone observed here (Supplementary Fig. 10), and in *M. smegmatis* menaquinone-9 molecules were observed occupying both sites[31]. Together, these observations suggest ubiquinone-10 can transit along the closed-state channel, resting part way in stable transit intermediates, such as that identified here.

### Water molecules and hydrated networks

Most of the water molecules assigned in *Pd*-CI-ND are in the hydrophilic domain, but there are also many in the membrane domain, where they may partake in proton pumping. Connections between Grotthuss-competent residues and waters were identified to evaluate candidate proton transfer routes[15,50,51], but keeping in mind that our structure is of an enzyme resting state, and that additional routes may form transiently in specific catalytic states, or be extended as

unobstructed gaps are crossed by thermal fluctuations or unresolved dynamic waters. Our analysis reveals a similar picture to that discussed recently in detail for complex I from mouse heart mitochondria[51].

At the top of the ubiquinone-binding site, where protons are required for ubiquinone reduction, NDUFS2-Tyr88 is hydrogen-bonded to just two ordered waters (Fig. 6a), but a substantial Grotthuss-competent network (separated from His39 by an unobstructed 4.6-Å gap) leads from NDUFS2-Asp404 almost to the solvent-accessible surface (route 1 in Fig. 6c and Supplementary Fig. 11a). Further down the channel, a network starting 4.8 Å from the bound ubiquinone headgroup, including ND1-Glu222, Glu224 and Glu226, connects the top of the E-channel to the surface at the domain corner (route 2 in Fig. 6c and Supplementary Fig. 11b). Based on their well-defined densities[52], the ND1-Asp219 (connected to Glu224) and Glu226 carboxylates are probably protonated (Supplementary Fig. 11c). ND1-Asp219 (*Mm*Asp199) was also proposed to be protonated in the closed state of mouse complex I[51] and molecular simulations in *Y. lipolytica* CI[32] have suggested protons are taken up to Asp219 from the N-side, then transferred to the P-side via the E-channel. An alternative route, not observed here, was also proposed in *Y. lipolytica* CI[53], to NDUFS2-His35 via ND3-Glu46 and ND1-Lys141. Furthermore, due to unobstructed network-gaps on both sides of NDUFS2-Asp140 we do not observe the complete network between the E-channel and NDUFS2-His39 that was previously observed in *E. coli* CI[15] (Figs. 6a and 6c).

Many structured waters are present in the *Pd*-CI E-channel, although a complete Grotthuss-competent proton-transfer pathway

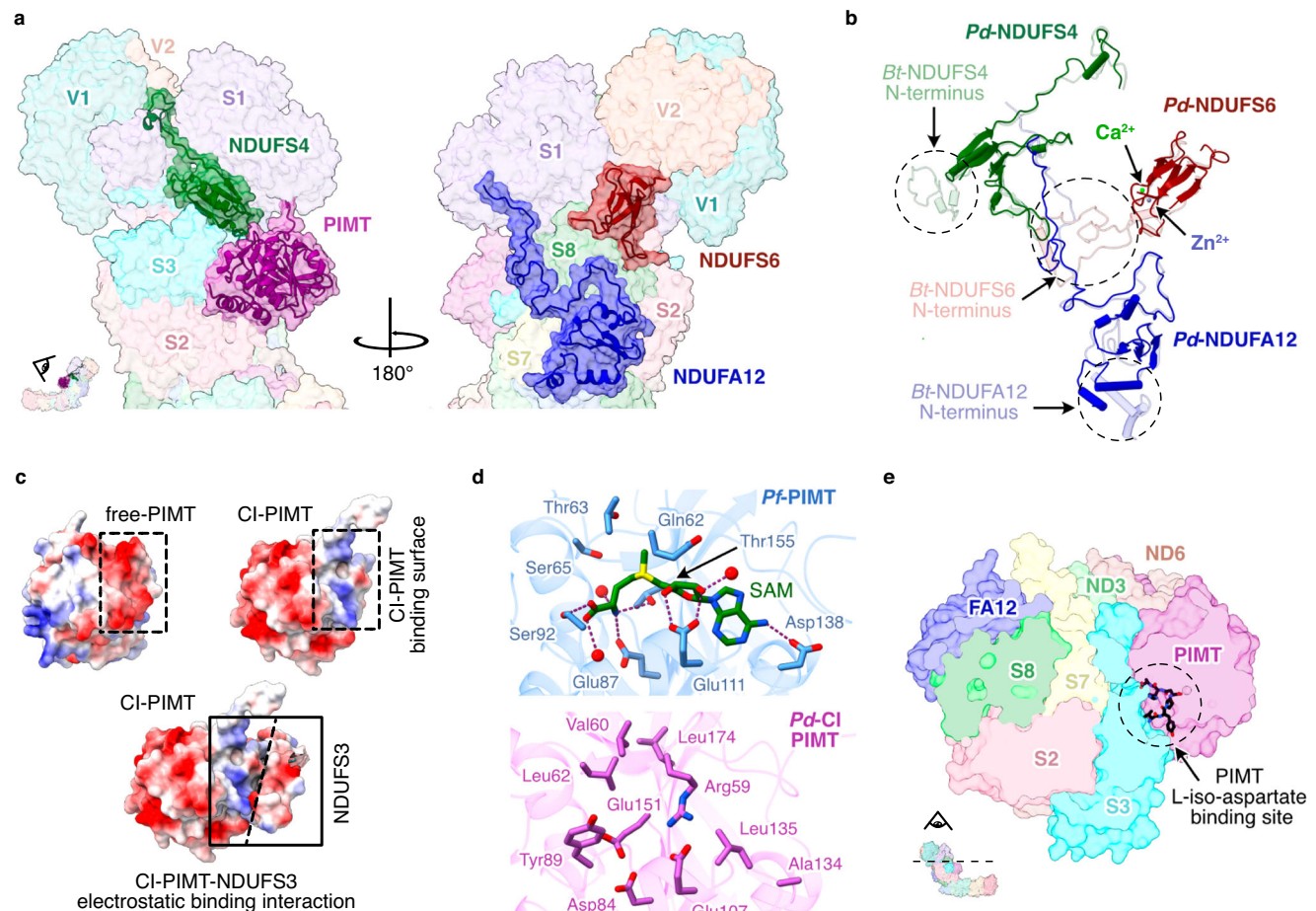

**Fig. 4 | Supernumerary subunits in *P. denitrificans* complex I. a** The locations of subunits NDUFS4 (green), CI-PIMT (purple), NDUFA12 (blue) and NDUFS6 (red). **b** Secondary structure comparison of subunits NDUFS4, NDUFS6 and NDUFA12 with bovine complex I PDB-7QSL[20], shown in their arrangement in the structure. **c** Comparison of electrostatic surface of free-PIMT and CI-PIMT showing binding of CI-PIMT is governed by electrostatic interactions with NDUFS3. **d** Comparison of key residues involved in cofactor and substrate binding in the functional PIMT from *Pyrococcus furiosus* PDB-1JG4[67] and CI-PIMT. **e** Cross-section of the *Pd*-CI hydrophilic domain showing that substrate binding in PIMT is obstructed in the complex.

was not evident (route 3 in Fig. 6c and Supplementary Fig. 11d), and along the central axis filling many of the gaps that are otherwise too distant for proton transfer (route 4 in Fig. 6d and Supplementary Fig. 11e). Although (as noted previously in mouse and *E. coli*[15,51]) all three TMH5-Glu(εO):TMH7-Lys(ζN) pairs are too far apart for ion-pair interactions, a complete Grotthuss network is observed within route 4 from ND4L-Glu73 to ND5-Lys238. No intact connections to the cytosol are observed, as potential proton uptake channels are broken at ND2-Tyr231, ND4-His246 and ND5-His263 (Fig. 6d), but may be expected to form during catalysis. An alternative rotamer of ND5-Met252, suggested by an unoccupied density, may allow connections to form more easily from ND5-His263 to the cytoplasm via Ser259 (Supplementary Fig. 11e). The 13 Å gap between ND5-Lys238 and ND5-His263 in route 4 is obstructed by several hydrophobic residues, including the putative gating residues Trp247 and Leu248[54], as observed previously in *E. coli* complex I[15], whereas in other structures, including the mouse[51] and bovine[20] enzymes, the histidine is observed in a different orientation, pointing toward Lys238. Following this gap, ND5-His263 is connected to ND5-Lys413 on the ND5-TMH12 loop above the periplasmic ND5 cavity, and Lys413 may connect to the periplasm via ND5-Asp414, His404 and Glu418, across a 5.6-Å unobstructed gap. No additional connections to the periplasm are observed, but our data do not exclude them forming transiently in different states during catalysis.

## Discussion

Our high-resolution structure of *Pd*-CI provides a powerful platform for combining structural, biophysical, biochemical and genetic strategies in any subunit within a single complex I model. Previously[11] we used mammalian structures to design site-directed mutants in *Pd*-ND4, a proton-pumping subunit that is mitochondrial-encoded in eukaryotes, and determined their rates of catalysis, proton-pumping stoichiometries, and structural integrities. Our direct structural analyses now enhance the potential of this and similar studies: variants that stop catalysis offer opportunities to trap intermediates, and variants proposed to hinder specific reaction steps may be investigated in more depth. For example, the ND4-H320L and H346Q mutations (between K263 and E405 in the central axis) showed slower catalysis but unaltered proton stoichiometry[11]. Structural data may now be used to interrogate the proposal from molecular dynamics simulations[55] that they are able to recruit water molecules to replace the hydrogen bonding provided by His sidechains. The combination of approaches that can be brought to bear on *Pd*-CI make it a uniquely powerful model system for studying the mechanism of complex I catalysis.

*Pd*-CI contains three mammalian-type supernumerary subunits (NDUFS4, NDUFS6 and NDUFA12) that predate eukaryogenesis[14,17], plus a PIMT subunit, so far specific to *Pd*-CI, that was incorporated following the endosymbiotic event. The three ancestral subunits appear to stabilise the N-module, with their roles fulfilled partly in *E.*

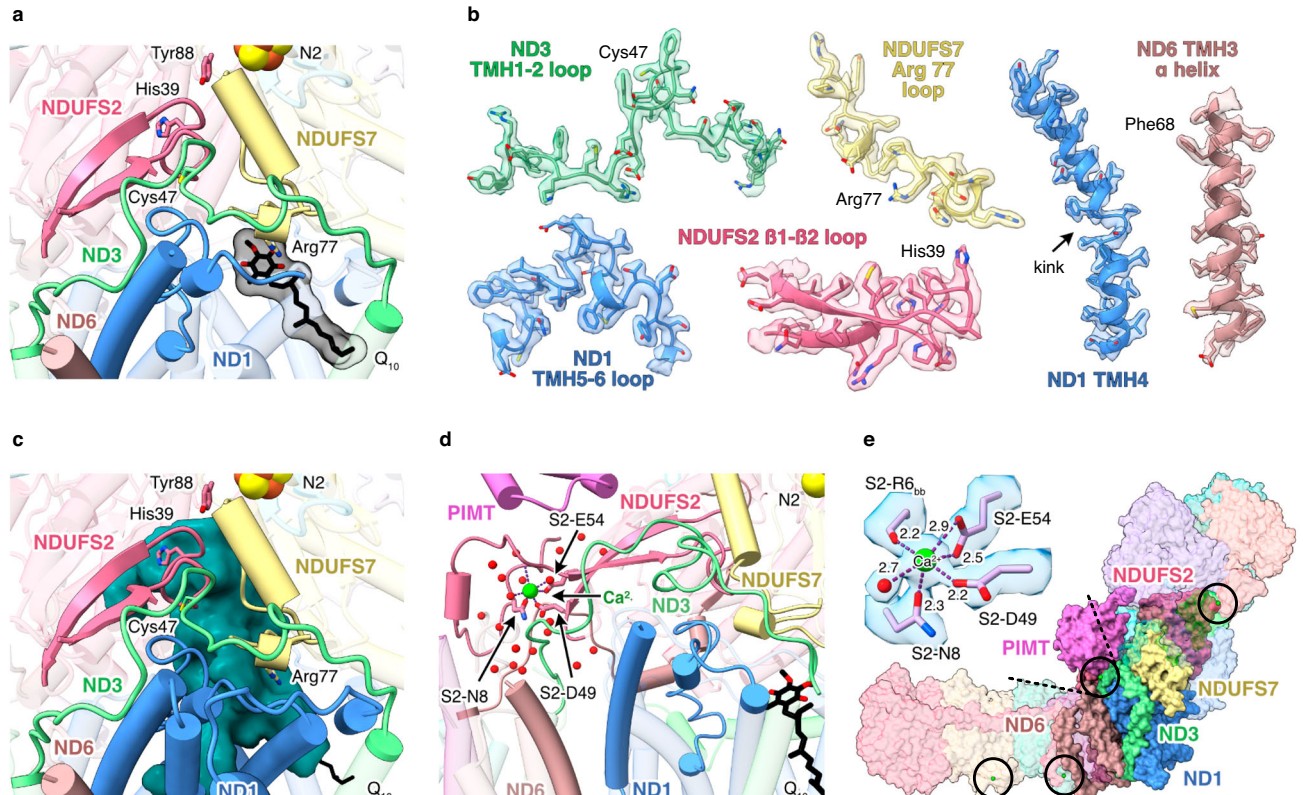

**Fig. 5 | P. denitrificans CI is in a homogeneous closed state. a** The conformations of the loops that form the ubiquinone-binding site and the location of the bound ubiquinone-10 in the lower section of the channel. **b** Cryo-EM densities for key structural elements that define the closed state. **c** The ubiquinone-binding site (determined by CASTp analysis) is closed and sealed from the cytoplasm (shown from the same orientation as panel **a**). **d** The corner between the hydrophilic and hydrophobic domains showing the N-terminus of NDUFS2 and the bound Ca²⁺ ion. **e** Coordination of the corner bound Ca²⁺ ion and the packing of surrounding subunits; locations of three additional bound Ca²⁺ are also indicated.

coli, *M. smegmatis* and *T. thermophilus* complexes I by extensions of the core subunits, demonstrating how different lineages adopt diverse stabilisation strategies[14]. *Pd*-NDUFS4 is substituted by a Ca²⁺-stabilised G-loop extension of *E. coli* NDUFS1[15,30], an N-terminal extension of *M. smegmatis* NDUFS3[31] and a shorter G-loop extension and subunit Nqo16 in *T. thermophilus*[29]. *Pd*-NDUFS6 is substituted partly by NDUFS8 in *E. coli* and *M. smegmatis*, and by the frataxin-like Nqo15 subunit in *T. thermophilus*. NDUFA12 is not substituted in any of the three species, suggesting that it is less important for structural integrity. NDUFS4, NDUFS6 and NDUFA12 are also proposed to co-operate in assembling the N-module onto the mammalian complex[38,56], although their absence from most prokaryotic enzymes suggests this role is not universal.

Strikingly, *Pd*-CI is observed resting entirely in the closed turnover-ready state. In contrast, mammalian complex I is typically observed resting in a mixture of closed/open states, the *E. coli* enzyme has been observed entirely in the open state, and the complexes from *Y. lipolytica* and *T. thermophilus*, the plants *Brassica oleracea* and *Arabidopsis thaliana*, and the bacterium *Thermosynechococcus vestitus*, in intermediate states with some elements closed-like and others open-like[45]. Only the complexes from the protozoan *Tetrahymena thermophila*[33] and *M. smegmatis*[31] have been found previously in homogeneous closed states. Here, we proposed the *Pd*-CI closed state is stabilised by the PIMT subunit and the N-terminus of NDUFS2, packed into the 'corner' between the hydrophilic and membrane domains and stabilised by a bound Ca²⁺. This Ca²⁺ ion may explain the dependence of *Pd*-CI catalysis and stability on Ca²⁺[12]. The *T. thermophila* enzyme was similarly proposed to be locked closed by an extended cohort of interconnected supernumerary subunits at the domain corner[33], and it also exhibits similar structures for

the NDUFS2 N-terminus and the ND6 C-terminal extension. In *Pd*-CI the ordered NDUFS2 N-terminus contacts PIMT, ND6 N-terminus and ND3 TMH1-2 loop, which changes its conformation when the mammalian enzyme opens (deactivates). The ND6 C-terminus is also extended in *M. smegmatis* complex I, where it is folded into the interdomain corner, interacts with a short insert in the ND3 TMH1-2 loop, and is stabilised by an extended ND2 TMH3-4 loop. Finally, the extra supernumerary subunit MSMEG_2064 in *M. smegmatis* complex I occupies the same place as the mammalian NDUFA9 subunit, which also changes conformation upon opening[13]. Together, the data suggest common strategies for stabilising the closed resting state: the packing of the interdomain corner and structures to lock the ND3 TMH1-2 loop into place.

The homogeneous closed resting states of *Pd*-CI and other species raise obvious questions. Is opening a regulatory mechanism that has evolved in some species but not in others? Could open states represent a 'safety net' state to catch the enzyme, and provide the opportunity to recover following a first unfolding step – whereas enzymes such as *Pd*-CI become committed to an irreversible downhill trajectory? Are these closed-only complexes locked permanently closed, even during turnover, or could opening be an energy-intensive activated process that only occurs during catalysis? The combination of structural, functional and genetics approaches now possible for *Pd*-CI, together with the clarity provided by an enzyme set in a single, homogeneous state before catalysis is initiated, provide attractive new opportunities for focussed and robust approaches to answering these questions.

## Methods
All chemicals and reagents were purchased from Merck unless stated otherwise.

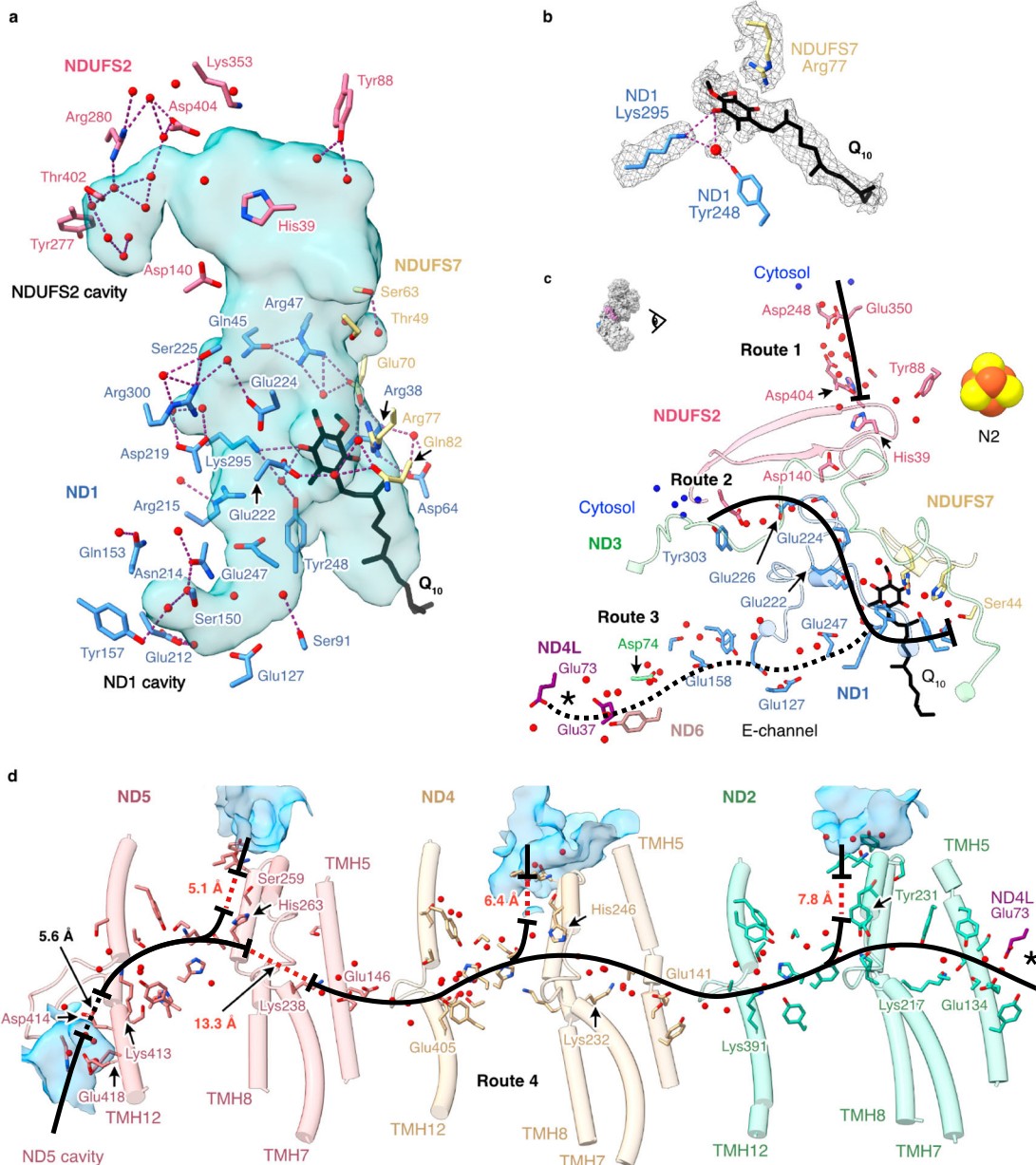

**Fig. 6 | Hydrogen bonding and Grotthuss competent network of the ubiquinone binding site, E-channel and central axis observed in the turnover-ready resting state of *P. denitrificans* complex I. a** Hydrogen bonding networks between protonatable residues, water molecules (red spheres) and ubiquinone-10 at the ubiquinone-binding channel identified using the *hbonds* command in UCSF ChimeraX. The cavity was determined using CASTp analysis with a 1.4 Å radius probe and comprises the substrate binding channel plus an extension into the core of ND1 and an additional lobe in NDUFS2. **b** Cryo-EM densities for the bound ubiquinone-10, neighbouring protein residues and water molecules. **c** Grotthuss-competent networks at the ubiquinone-10 binding site (routes 1, 2 and 3), depicting potential ubiquinone-10 protonation routes, and the discontinuous Grotthuss competent network in the E-channel (route 3). **d** The central axis of ND2, ND4 and ND5 subunits in *Pd*-CI (route 4) with solvent accessible cavities in blue. In **c**, **d**, solid black lines indicate continuous Grotthuss competent networks, while dotted black and red lines denote unobstructed and obstructed gaps in the networks, respectively. Solvent-exposed water molecules are in blue. The asterisks in **c** and **d** indicate where the two pathways connect. Detailed views of the Grotthuss networks in **c** and **d** are presented in Supplementary Fig. 11a, b, d, e.

## Preparation of complex I from *P. denitrificans*

*P. denitrificans* membranes were prepared according to a protocol adapted from Jarman et al. [12] 12 × 2 L non-baffled Erlenmeyer flasks containing 500 mL of fresh LB broth were inoculated with 0.1% (v/v) of overnight pre-culture of the *Pd*-Nqo5[His6] strain[12]. Cells were grown aerobically at 30 °C with 225 rpm shaking and harvested at late-log phase (OD$_{600}$ 4–4.5) by centrifugation (5000×*g*, 30 min, 4 °C). All subsequent steps were performed at 4 °C. Cells were resuspended and homogenised in *ca.* 2.5 mL of lysis buffer (50 mM MES, pH 6.5 at 4 °C, plus one cOmplete™ EDTA-free protease inhibitor cocktail tablet

(Roche) per 50 mL) per gram of wet cell pellet, then lysed using a Z-plus 2.2 kW cell disruptor (Constant Systems Limited), once at 15,000 PSI and twice at 25,000 PSI. Cell debris was removed by centrifugation (31,900 × *g*, 1 h) then membranes harvested by ultracentrifugation (234,800 ×*g*, 2 h), resuspended in 50 mM MES buffer (pH 6.5 at 4 °C), flash frozen in liquid nitrogen (LN$_2$) and stored at −70 °C.

Complex I for the *Pd*-CI-DDM cryo-EM analysis was prepared as described by Jarman et al. [12] except that glycerol was omitted from the final size-exclusion chromatography buffer. Complex I for the *Pd*-CI-

ND analysis was prepared using the same protocol with minor modifications. Membranes (~500 mg protein) were thawed and adjusted to 10 mg mL$^{-1}$ in buffer containing 20 mM MES (pH 6.5 at 4 °C), 100 mM NaCl, 5 mM CaCl$_2$, 20 mM imidazole, 10% (v/v) glycerol, and the cOmplete™ EDTA-free protease inhibitor cocktail. They were solubilised by dropwise addition of $n$-dodecyl-β-D-maltoside (DDM, Anatrace) to a final concentration of 2.85%, and stirred for 30 min. Insoluble material was removed by centrifugation (172,000 × $g$, 45 min), then the supernatant was passed through a 0.45 µm syringe filter (Millex$^{HP}$, Millipore) and loaded onto a 5 mL His-Trap™ HP column (Cytiva) equilibrated with buffer A (20 mM MES (pH 6.5 at 4 °C), 150 mM NaCl, 2.5 mM CaCl$_2$, 80 mM imidazole, 10% (v/v) glycerol, 0.1% (w/v) DDM, 0.02 % (w/v) soy bean asolectin (Avanti Polar Lipids) and 0.02% (w/v) 3-[(3-cholamidopropyl)dimethylammonio]-1-propane-sulfonate (CHAPS). The column was washed with buffer A until the A$_{280}$ absorbance dropped to its baseline, then complex I was eluted in buffer B (buffer A containing 300 mM imidazole). Complex I-containing fractions were pooled, concentrated to 1 mL (100 kDa MW cut-off, Merck Millipore), filtered, and loaded onto a Superdex 200 Increase 10/300 GL size exclusion column (Cytiva), equilibrated with buffer C (20 mM MES (pH 6.5 at 4 °C), 150 mM NaCl, 2.5 mM CaCl$_2$, 10% (v/v) glycerol and 0.05% (w/v) DDM). Complex I-containing fractions were pooled and concentrated to 15–20 mg mL$^{-1}$, before glycerol was added to a final concentration of 30% (v/v) and the protein was flash frozen in LN$_2$ and stored at −70 °C.

## Preparation of MSP2N2

MSP2N2 was prepared according to a protocol adapted from Chung et al.[20] 2 L of LB broth were inoculated with 1% (v/v) of overnight culture of *E. coli* strain NiCo21(DE3) (Dr Ali Ryan, Northumbria University) containing pMSP2N2 (Addgene), and cells grown aerobically at 37 °C with 225 rpm shaking to mid-exponential phase (OD$_{600}$ ~ 0.6). Protein expression was induced by 0.1 mM isopropyl β-d-1-thiogalactopyranoside (IPTG), then after 4 h the cells were harvested by centrifugation (6000×$g$, 20 min), resuspended in lysis buffer (50 mM Tris-HCl (pH 8), 500 mM NaCl, 5% glycerol, 1% v/v Triton X-100, 0.002% (w/v) PMSF, 1 mM EDTA, 5 mM MgCl$_2$, and 1 EDTA-free protease inhibitor tablet (Roche) per 100 mL) and frozen. For purification of MSP2N2, cell suspensions were thawed, a few crystals of DNase I were added, then the cells were lysed by sonication on ice using a Qsonica sonicator (50% power, 30 cycles of 10 s on, 20 s off). The lysate was clarified by centrifugation (30,000×$g$, 1 h, 4 °C), filtered, and applied to a 1 mL His Trap HP column (Cytiva) equilibrated with buffer A (50 mM Tris- HCl (pH 8), 500 mM NaCl) plus 1% v/v Triton X-100. The column was washed with 5–10 mL of buffer A plus 1% v/v Triton X-100, 5–10 mL of buffer A plus 50 mM sodium cholate, 5–10 mL of buffer A plus 20 mM imidazole, 3–5 mL of buffer A plus 80 mM imidazole, then MSP2N2s were eluted with buffer A plus 500 mM imidazole. The eluate was analysed via SDS-PAGE, then MSP2N2-containing fractions were reapplied to a 5 mL HisTrap column (HP, Cytiva) and the procedure was repeated. MSP2N2 fractions were collected and dialysed against 5 L of 10 mM MOPS-KOH pH 7.5, 50 mM KCl, before the pure MSP2N2 was concentrated, flash frozen in LN$_2$ and stored at −70 °C.

## Reconstitution of *Pd*-CI into phospholipid nanodiscs (*Pd*-CI-NDs)

Nanodiscs were prepared using a protocol adapted from Chung et al.[20] 0.5 mg of lipids (8:1:1 mixture of 1,2-dioleoyl-*sn*-glycero-3-phosphocholine (PC), 1,2-dioleoyl-sn-glycero-3-phosphoethanolamine (PE), and 18:1 cardiolipin (CDL), Avanti Polar Lipids) in chloroform were combined with Q$_{10}$ (20 nmol (mg lipid)$^{-1}$), then placed under a N$_2$ stream to remove the solvent and dried under vacuum for 2 h. Then, the lipids/Q$_{10}$ were hydrated in 0.5 mL of reconstitution buffer (20 mM MES (pH 6.5), 25 mM NaCl) for 30 min with frequent vortexing. Sodium cholate was added to 40 mM, and the mixture sonicated in a bath

sonicator for 10 min. All subsequent steps were performed at 4 °C. MSP2N2 and *Pd*-CI were added to the lipid/Q$_{10}$ mixture at a molar ratio of 400 lipids:10 MSP2N2:1 *Pd*-CI, mixed, and diluted to a volume of 1 mL. The mixture was incubated for 20 min on ice, then sodium cholate was removed using a PD10 desalting column (Cytiva). The nanodiscs were concentrated to 100 µL (centrifugal 100 kDa MW cut-off concentrator, Merck Millipore) then eluted from a Superose 6 Increase 5/150 size exclusion column on an Akta Micro (Cytiva). The total protein (*Pd*-CI and MSP2N2) concentration was determined using the Pierce™ bicinchoninic acid (BCA) assay (Thermo Fisher Scientific).

## Kinetic measurements

All kinetic assays were carried out at 32 °C in a Molecular Devices SpectraMax 348 96-well plate reader in buffer containing 10 mM MES (pH 6.5 at 32 °C), 25 mM NaCl and 2 mM CaCl$_2$. The *Pd*-CI content in *Pd*-CI-NDs was quantified by comparison to a standard sample of *Pd*-CI in DDM using the NADH:APAD$^+$ oxidoreduction assay (where APAD$^+$ is 3-acetylpyridine adenine dinucleotide) with 100 µM NADH, 500 µM APAD$^+$, 1 µM piericidin A and 0.1% DDM, monitored using $\varepsilon_{(400-450 nm)} = 3.16$ mM$^{-1}$ cm$^{-1}$. NADH:O$_2$ oxidoreduction by membranes was measured using 200 µM NADH and 12.5 µg mL$^{-1}$ alamethicin with $\varepsilon_{(340-380 nm)} = 4.81$ mM$^{-1}$ cm$^{-1}$. NADH:O$_2$ oxidoreduction by *Pd*-CI-ND was measured using 200 µM NADH and 10 µg mL$^{-1}$ AOX[23,57]. NADH:decylubiquinone (dQ) oxidoreductase activity was typically measured in 200 µM NADH, 200 µM dQ, 0.15% soy bean asolectin and 0.15% CHAPS.

## Nano-DSF

*Pd*-CI-DDM and *Pd*-CI-ND were diluted to ~0.2 mg mL$^{-1}$ in the appropriate buffer, loaded into capillaries, then placed in a Prometheus NT.48 instrument (NanoTemper Technologies). The temperature was raised from 20 to 90 °C at 4.5 °C min$^{-1}$ and the tryptophan fluorescence monitored at 330 and 350 nm using an excitation wavelength of 280 nm.

## Cryo-EM grid preparation, data acquisition and processing

UltrAuFoil 0.6/1 grids (Quantifoil Micro Tools GmbH, Germany) were prepared as described previously[44]. Briefly, grids were glow discharged (20 mA for 90 s), incubated under anaerobic conditions in 5 mM 11-mercaptoundecyl hexaethylene glycol (TH 001–m11.n6–0.01, ProChimia Surfaces) in ethanol for 48 hr, then washed several times with 100% ethanol and air dried. 2.5 µL of 3 mg-protein mL$^{-1}$ *Pd*-CI-DDM, 2.1 mg-protein mL$^{-1}$ *Pd*-CI-NDs (without FOM) or 1.9 mg-protein mL$^{-1}$ *Pd*-CI-NDs (with FOM) were applied to each grid in an FEI Vitrobot IV (Thermo Fisher Scientific) at 4 °C and 100% relative humidity before blotting for 10 s with a blot force setting of -10 and vitrifying the sample in liquid ethane. Grids were screened on a 200 keV FEI Talos Arctica microscope at the cryo-EM facility at the Department of Biochemistry, University of Cambridge. Data for the *Pd*-CI-DDM dataset, and for *Pd*-CI-ND grids 1 & 2 were collected at the University of Cambridge facility, and data from grid 3 at the UK National Electron Bio-Imaging Centre (eBIC) beamline M07 at the Diamond Light Source (Harwell Science and Innovation Campus, Didcot, UK). The *Pd*-CI-DDM dataset was imaged using a Gatan K2 detector and energy filter (Gatan BioQuantum), and the *Pd*-CI-ND data using a Gatan K3 detector and energy filter (Gatan BioContinuum). In all cases, the energy filter was operated in zero-loss mode with a slit width of 20 eV, on a 300 keV FEI Titan Krios microscope (Thermo Fisher Scientific) with a 100 µm objective aperture and C2 apertures of 50 and 70 µm for the *Pd*-CI-DDM and *Pd*-CI-ND data, respectively and data were recorded using EPU software. *Pd*-CI-DDM images were recorded in counting mode at nominal magnification of 130,000×, giving a facility-provided pixel size of 1.07 Å pixel$^{-1}$ (later calibrated to 1.05 Å pixel$^{-1}$) with a defocus range of -1.5 to -2.9 µm in 0.3 µm increments and the autofocus routine run every 10 µm; the dose rate was 4.8 $e^-$ Å$^{-2}$ s$^{-1}$ with 10 s of exposure, giving a total dose of 48 $e^-$ Å$^{-2}$ captured in 25 frames; there was a delay

of 7 s after stage shift and 3 s after image shift. *Pd*-CI-ND images were collected in super-resolution mode without hardware binning at a nominal magnification of 81,000×, giving facility-provided pixel sizes of 1.066 (grids 1 and 2) and 1.06 (grid 3) Å pixel$^{-1}$, with defocus ranges of −1.0 to −2.4 (grids 1 and 2) and −0.9 to −2.5 (grid 3) μm in 0.2 μm increments and the autofocus routine run every 10 μm. The dose rates were 19.43 (grids 1 and 2) and 11.25 (grid 3) $e^{-}$ Å$^{-2}$ s$^{-1}$ with 2.4 (grids 1 and 2) and 4.0 (grid 3) s exposures giving total doses of 46.6 (grids 1 and 2) and 45 (grid 3) $e^{-}$ Å$^{-2}$ captured in 40 frames. All data were acquired as one shot per hole in aberration-free image shift (AFIS) mode with 5 s delay after stage shift and 1 s delay after image shift.

## Cryo-EM image processing for *Pd*-CI-DDM

Data for *Pd*-CI-DDM were first processed using RELION 3.0.7[58] (Supplementary Fig. 1a). Micrograph motion was corrected for with RELION's implementation of motion correction with 5 × 5 patches, and the contrast transfer function (CTF) was estimated using GCTF v 1.18[59] with an amplitude contrast of 0.1. Ice-contaminated micrographs were removed manually, resulting in 2,040 micrographs, then 128,270 particles were picked using RELION's AutoPicking tool with a 2D map reference generated from a set of manually picked particles. The particles were extracted at 1.07 Å pixel$^{-1}$ and subjected to initial 2D and 3D global classifications. The final 3D class, containing 19,595 particles, was used as a 3D model to repick 125,639 particles from all micrographs and repeat the pipeline. The final 3D class of 51,308 particles was subjected to iterative 3D and CTF refinements and Bayesian polishing to produce a final map of 4.1 Å global resolution at FSC = 0.143 that showed poor density for the distal section of the membrane domain. The particles were imported into CryoSPARC v 3.3.2[60], homogenously refined and subjected to 3DVA analyses[61]. The analysis uncovered a large movement of the membrane domain, and the volume frames at the extremes of the motion were used as inputs to 3D classification to yield two distinct classes that were subjected to NU refinement: class 1 (19,736 particles, 4.5 Å global resolution) and class 2 (31,572 particles and 4.2 Å global resolution) (Supplementary Figs. 1b and 2a, b).

## Cryo-EM image processing for *Pd*-CI-ND

The *Pd*-CI-ND datasets were first processed separately, mostly using RELION 3.1.0[58] but with some steps undertaken in CryoSPARC v. 2.7 or 3.3.2[60] (Supplementary Figs. 5 and 6). RELION's implementation of motion correction was used with 2 × binning and 5 × 5 patches, and the CTF estimated using CTFFIND-4.1[62] with an amplitude contrast of 0.1 and maximum resolution of 4 Å. Following filtering to remove micrographs with CtfFigureOfMerit <-0.05, MaxResolution >7.5 Å, and CtfAstigmatism <25 or >1000, ice-contaminated micrographs were removed manually. 846,255, 188,249 and 320,429 particles were then picked from grids 1, 2, and 3, respectively, using RELION's AutoPicking tool using 2D map references of bovine complex I. The particles were extracted with a box size of 450 pixels, downscaled to 200 pixels, and subjected to initial 2D and 3D global classifications to remove junk particles. Particles from grid 1 were 2D classified in cryoSPARC yielding 814,431 particles, and particles from grid 2 were 2D classified using RELION yielding 184,540 particles. As <5% of the particles were eliminated, 2D classification was not used for grid 3. 3D classification on grid 1 was carried out by ab-initio reconstruction and heterogeneous refinement in CryoSPARC, and for grids 2 and 3 in RELION, using global classification with searches to 3.7°, yielding 324,569, 66,267 and 255,764 particles for grids 1, 2 and 3, respectively. The particles were re-extracted in RELION at box size of 450 pixels, and the particles for grids 1 and 2 were combined. Then, the particles were 3D refined and 3D classified using local angular searches at 3.7, 1.8 and 0.9° resulting in 181,518 particles from grids 1-2, and 71,764 particles from grid 3. Poor density was noted for the distal section of the membrane domain so focussed 3D classifications without angular searches were performed

using a low-pass filtered 15 Å-resolution mask with a soft edge of 10 pixels, which was created from a map generated using the *molmap* function in UCSF ChimeraX[63] from a model of the ND5 subunit of bovine complex I. The resulting 103,186 and 43,417 particles of intact complex I from grids 1-2 and 3, respectively, were subjected to iterative 3D refinements and CTF refinements (for anisotropic magnification, beam-tilt, trefoil, and per particle defocus, astigmatism and *B*-factors). Particle motion was further corrected using Bayesian polishing, during which the particle stacks were extracted with a box size of 900 and rescaled to 640 pixels for grids 1-2 and 3, respectively (giving pixel sizes of 0.749 and 0.745 Å pixel$^{-1}$). Further 3D and CTF refinements then produced two maps with resolutions of 2.4 and 2.5 Å that exhibited a map-map correlation of 0.99 in ChimeraX, indicating they are essentially identical.

To merge the datasets (Supplementary Fig. 7), the pixel sizes in the stacks needed to match very closely and it was found necessary to re-run motion correction for grids 1 and 2 with a corrected pixel size of 1.068 Å. The final particles for grids 1 + 2 were then extracted using a box size of 894 pixels and rescaled to 640 pixels, corresponding to 0.745 Å pixel$^{-1}$ (that of grid-3), 3D and CTF refined, and polished. Then, the particles from all three grids were merged and subjected to a final cycle of iterative 3D and CTF refinements. For global protein masks (excluding MSP2N2 signal), low-pass filtered 15 Å-resolution masks with a soft edge of 10 pixels were produced in RELION's mask create tool from a map generated by *molmap* from a near-complete model. Final 3D refinement with solvent flattening produced a map with a global resolution of 2.3 Å at FSC = 0.143 (Supplementary Fig. 2c). The density for the distal section of the membrane domain was refined separately. A 3D refinement without alignment (--skip_align) was performed to restore previously masked areas, then the signal from subunits ND4-ND5 was isolated by subtraction, using a mask created from the consensus model. The subtracted particles were 3D auto-refined with default settings, resulting in a local map of improved quality with a global resolution of 2.5 Å (Supplementary Fig. 2d).

The 2.3-Å resolution consensus map of complex I was further processed to investigate unassigned fragmented densities (Supplementary Fig. 8). The particles were imported into CryoSPARC v 3.3.2[60] and homogeneously refined using the final map from RELION as a 3D model. 50 K of the particles were then analysed by 3DVA[61] with three components and a filter resolution of 10 Å. 20 output volumes per component from 3D variability display in simple mode were visualised in UCSF ChimeraX[63], and the particles for component 2 split into five groups using 3D variability display in intermediates mode. The first and last groups contained 2.3 K and 2.2 K particles, respectively, that were homogeneous refined separately. The volumes were then used as 3D models in a heterogenous refinement of all the particles, which split them into 100,233 particles with a 'tight' nanodisc and 46,370 particles with an extended nanodisc-like density. The first class was subjected to homogeneous refinement and focused 3D classification on a large, fragmented density above the distal membrane domain, identifying 38,555 particles with the additional density. The signal for the additional density was isolated by subtraction and reconstructed homogenously into a 3D volume, which served as a reference to refine the particles using Local Refinement in CryoSPARC, producing a final map with a global resolution of 3.4 Å.

## Model building for *Pd*-CI-ND

The sequences for known *Pd*-CI subunits[12,17] were input to ModelAngelo[26] to build a *de-novo* model from the 2.3-Å resolution consensus map. The initial coordinate model did not include any of the cofactors or ligands, and so the expected 2 x [2Fe-2S] and 6 x [4Fe-4S] clusters, flavin mononucleotide, and $Zn^{2+}$ were added manually, along with their co-ordinating protein ligands where necessary. The model was all-atom refined in Coot 0.9[64] and visually inspected. Additional density was observed next to the Met residue at the *N*-terminus of

subunit ND3 and modelled as formyl-Met. Additional density was also found next to the *N*-terminus of subunit ND6, which was found to begin with the genomic sequence ATGATG at the start; the Uniprot sequence assumed the second Met to be the initiator and adding a further Met explained the additional density. Density matching dimethyl arginine was modelled for NDUFS2-Arg65[36,37], and phospholipid and ubiquinone-10 densities were manually assigned, built and clipped. Where possible, phospholipids (particularly cardiolipin) were assigned from the cryo-EM densities and head-group/protein interactions, taking account of the composition of the reconstitution mixture (PC, PE and CDL) and of native *P. denitrificans* membranes (additionally phosphatidyl-glycerol (PG), phosphatidyl-serine (PS) and phosphatidic acid (PA))[65]. PC and PE were distinguished by the bulkier PC head group and its propensity to associate with aromatic side chains. Phospholipids with poor head group densities were built as PA. Two PS were built into regions with high likelihood of retention during purification, one behind the ND5 transverse helix and one in a lipid-rich cleft formed by ND1, NDUFS8 and NDUFA12. Rounds of real-space-refinement in Phenix v 1.20[66] and manual refinement in Coot were completed, with additional checks using Q-score analyses in UCSF ChimeraX to identify suspect backbone flips and rotameric positions. For real-space-refinement, a custom CIF restraint file was generated in Phenix for ubiquonone-10 to enforce the planarity of the ring. Five cycles of automated real-space-refinement was performed including global minimisation, local grid search, NHQ flips and ADP refinement, using Oldfield Ramachandran restraints for Favoured and Allowed, and Emsley8k for Outliers. Water molecules were added using the Coot *add waters* function and checked manually to remove any with poor density or coordination. Where solvent molecules appeared to have octahedral coordination, they were assigned as $Na^+$ for relatively weak densities, or as $Ca^{2+}$ for densities noticeably stronger than nearby water densities. To identify Grotthuss competent networks (with the cut-off set to 4.5 Å here to allow for thermal fluctuations) the following command was used in ChimeraX:

sel #[model]/[chain]:Lys,Glu,Asp,His,Tyr,Thr,Ser | #[model]:HOH; sel sel&@N*@O*; sel sel&-mainchain; contacts sel restrict sel reveal true distanceOnly 4.5 dashes 1.

### Reporting summary

Further information on research design is available in the Nature Portfolio Reporting Summary linked to this article.

## Data availability

Structural data generated in this study have been deposited in the EMDB and PDB databases under the following accession codes: EMD-18324 and 8QBY (complex I-nanodiscs), EMD-18325 and 8QC1 (ND4&5 focussed refined map & model), EMD-19975 (https://www.ebi.ac.uk/emdb/EMD-19975, complex I-DDM Class 1) EMD-19976 (https://www.ebi.ac.uk/emdb/EMD-19976, complex I-DDM Class 2), EMD-19977 (https://www.ebi.ac.uk/emdb/EMD-19977, cytochrome $bc_1$ complex). The cryo-EM raw images are available from EMPIAR with the accession codes EMPIAR-12077 (complex I-nanodiscs grids 1 and 2), EMPIAR-12078, (complex I-nanodiscs grid 3), and EMPIAR-12079 (complex I-DDM). The functional data in Supplementary Figs 3 and 4 are provided in the Source Data file. Source data are provided with this paper.

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

## Acknowledgements

We thank I. Chung and Z. Yin for assistance with cryo-EM grid preparation, screening and data collections, J. J. Wright for carrying out the activity assays in Supplementary Fig. 4 and D. N. Grba for helping with cryo-EM data processing and modelling. We thank D. Chirgadze, S. W. Hardwick and L. Cooper (University of Cambridge Cryo-EM facility), and K. Morris (eBIC, Diamond Light Source) for assistance with grid screening and cryo-EM data collection during the COVID-19 pandemic. This work was supported by the Medical Research Council (MC_UU_00015/2 to J.H.).

## Author contributions

B.S.I. purified *Pd*-CI and MSP2N2, conducted nanodisc reconstitution and nano-DSF experiments, optimised cryo-EM grid preparation and data collection, processed cryo-EM data, performed structure model building, analyses and interpretations and prepared figures. H.R.B. advised on cryo-EM data collection, collected cryo-EM data for grid 3, and contributed to processing, model building, and structure inter-pretations. O.D.J. purified the initial *Pd*-CI-DDM sample, prepared cryo-EM grids and processed the data with contributions from H.R.B. J.H. acquired funding, initiated and supervised the project and contributed to the interpretation of the data. B.S.I and J.H wrote the manuscript with input from all authors.

## Competing interests

The authors declare no competing interests.
