## [Transparent Peer Review file · Nature Communications]

Structure of the turnover-ready state of an ancestral respiratory complex I

Corresponding Author: Professor Judy Hirst

This manuscript has been previously reviewed at another journal. This document only contains reviewer comments, rebuttal and decision letters for versions considered at Nature Communications.

Version 0:

Reviewer comments:

Reviewer #1

(Remarks to the Author)

The authors have addressed my concerns in their response letter and made appropriate changes in the text. After reading the manuscript again, I would like to raise one more point related to Fig. 6 which shows different putative proton transfer pathways. Route 4 appears as a connection between the E channel and the prominent P side exit path in ND5. There is an ongoing discussion about possible P side pathways in ND4 and ND2. The figure suggests that the authors envision a continuous path along the complete length of the membrane arm and not the more traditional view of proton transfer pathways in each of the antiporter like subunits ND5,4 and 2. I think it would be good to clarify your view on this topic to avoid misunderstanding the figure.

Reviewer #2

(Remarks to the Author)

In the revised manuscript, the authors addressed all my questions. For me, the manuscript can be published as is.
